# In Silico Identification of Anti-SARS-CoV-2 Medicinal Plants Using Cheminformatics and Machine Learning

**DOI:** 10.3390/molecules28010208

**Published:** 2022-12-26

**Authors:** Jihao Liang, Yang Zheng, Xin Tong, Naixue Yang, Shaoxing Dai

**Affiliations:** 1State Key Laboratory of Primate Biomedical Research, Institute of Primate Translational Medicine, Kunming University of Science and Technology, Kunming 650500, China; 2Yunnan Key Laboratory of Primate Biomedical Research, Kunming 650500, China

**Keywords:** SARS-CoV-2, machine learning, Traditional Chinese medicine

## Abstract

Severe acute respiratory syndrome coronavirus 2 (SARS-CoV-2), the causative pathogen of COVID-19, is spreading rapidly and has caused hundreds of millions of infections and millions of deaths worldwide. Due to the lack of specific vaccines and effective treatments for COVID-19, there is an urgent need to identify effective drugs. Traditional Chinese medicine (TCM) is a valuable resource for identifying novel anti-SARS-CoV-2 drugs based on the important contribution of TCM and its potential benefits in COVID-19 treatment. Herein, we aimed to discover novel anti-SARS-CoV-2 compounds and medicinal plants from TCM by establishing a prediction method of anti-SARS-CoV-2 activity using machine learning methods. We first constructed a benchmark dataset from anti-SARS-CoV-2 bioactivity data collected from the ChEMBL database. Then, we established random forest (RF) and support vector machine (SVM) models that both achieved satisfactory predictive performance with AUC values of 0.90. By using this method, a total of 1011 active anti-SARS-CoV-2 compounds were predicted from the TCMSP database. Among these compounds, six compounds with highly potent activity were confirmed in the anti-SARS-CoV-2 experiments. The molecular fingerprint similarity analysis revealed that only 24 of the 1011 compounds have high similarity to the FDA-approved antiviral drugs, indicating that most of the compounds were structurally novel. Based on the predicted anti-SARS-CoV-2 compounds, we identified 74 anti-SARS-CoV-2 medicinal plants through enrichment analysis. The 74 plants are widely distributed in 68 genera and 43 families, 14 of which belong to antipyretic detoxicate plants. In summary, this study provided several medicinal plants with potential anti-SARS-CoV-2 activity, which offer an attractive starting point and a broader scope to mine for potentially novel anti-SARS-CoV-2 drugs.

## 1. Introduction

Coronavirus disease 2019 (COVID-19), which is caused by severe acute respiratory syndrome coronavirus 2 (SARS-CoV-2), is characterized by rapid spread and strong infectivity [1]. Patients with COVID-19 will experience persistent symptoms including fever, dry cough, fatigue, or dyspnea. Moreover, severe cases can cause a variety of complications leading to death [2]. COVID-19 is spreading and has caused hundreds of millions of infections and millions of deaths worldwide [3]. Currently, three drugs have been approved by the Food and Drug Administration (FDA) for clinical use in the treatment of COVID-19. These drugs all directly target essential proteins of the SARS-CoV-2 virus. Remdesivir and Molnupiravir target RNA-dependent RNA polymerase (RdRp), and they belong to the nucleotide analog. Paxlovid targets 3-chymotrypsin-like protease (3CLpro) [4,5,6]. However, SARS-CoV-2 has strong variability and many variants have appeared, such as the Alpha variant, the Beta variant, the Gamma variant, the Delta variant, and the Omicron variant [7,8]. More seriously, the long-term use of antiviral drugs that target viral proteins may lead to the emergence of drug-resistant virus variants and the decline of antiviral drug sensitivity [9,10,11]. Therefore, with the continuous variation of SARS-CoV-2 and the rapid increase in the number of infections, there is an urgent need to discover more effective drugs for the further treatment of COVID-19.

It is well known that natural products have been regarded as the main sources of drug discovery. Newman et al. reported that 65% of the total 1211 new small-molecule drugs approved between 1981 and 2014 are directly or indirectly derived from natural products [12]. TCM has been used to treat various diseases for thousands of years in China, including viral diseases. TCM has been used in China since the first days of the COVID-19 pandemic outbreak. Shu Feng Jie Du and Lianhuaqingwen Capsule have been recommended as treatment due to their demonstrated efficacy against previous influenza A (H1N1) or SARS [13]. Moreover, to treat COVID-19, experts recommended the use of different kinds of TCM according to the disease stage. Additionally, some TCMs prevented the SARS-CoV-2 infection of healthy persons and improved the health of patients with mild or severe symptoms [14,15].TCMs, which contain multiple active ingredients that can exert anti-inflammatory and immunomodulatory effects through targeting multiple key targets and pathways, have shown a good therapeutic effect in the fight against the COVID-19 pandemic [16,17]. Many antiviral bioactive ingredients come from medicinal plants [18,19,20,21]. It has been reported that licorice and scutellaria baicalensis are the most frequently used Chinese herbs. Glycyrrhizin isolated from liquorice root and baicalin extracted from scutellaria baicalensis can effectively inhibit the replication of clinical isolates of SARS and SARS-CoV-2 virus [22,23,24,25]. Cepharanthine is an alkaloid from stephania japonica, a raw material of some TCMs. Its anti-SARS-CoV-2 ability has been found and verified in vitro and in vivo, and it has entered clinical trials [26]. Compared with synthetic drug-like molecules, natural products from TCM show a wide range of biological activities and structural diversity [27]. Natural products are invaluable sources for drug discovery and provide rich chemical inspiration for developing antiviral drugs. Screening natural ingredients or compounds from TCM may be an effective way to develop new drugs [28,29].

Despite the urgent need, it is unrealistic to develop new drugs within a short period. Traditional drug discovery requires a very long period of investment and that has so far shown a lower success rate [30]. In the emergency of the SARS-CoV-2 outbreak, virtual screening provides a fast and effective strategy to identify potential and novel antiviral compounds for optimization. At present, the screening of TCMs mainly uses molecular docking to target viral proteins, such as spike protein, nucleocapsid protein, RdRp, 3CLpro (M^pro^), and 2′-O-ribose methyltransferase [31,32,33,34,35]. YangYi et al. screened 125 small molecules from licorice through the molecular docking to the receptor binding domain (RBD) of the SARS-CoV-2 spike protein. They found that licorice-saponin A3 (A3) and glycyrrhetinic acid (GA) have high affinity for the RBD structure, and potently inhibit SARS-CoV-2 infection in cellular experiments, with an EC50 of 75 nM and 3.17 µM, respectively [36]. This method requires the 3D structure of the target protein and uses a lot of computing time to predict the position, direction and conformation when the ligand binds to the target protein. Another TCM screening method is based on cell-based screening, which also requires a lot of time [37,38]. Machine learning methods are widely applied in almost all stages of the discovery and development of new drugs [39]. There are many studies on SARS-CoV-2 using machine learning methods [40,41,42,43]. Moreover, virtual screening using machine learning methods is far more efficient than traditional drug discovery processes [44]. Some studies have adopted machine learning or deep learning methods to deal with the COVID-19 pandemic, especially aimed at repurposing existing drugs to treat the disease. They will be divided into two main groups: target-based and phenotype-based drug discovery methods [45,46,47,48,49]. For example, Ivanov et al. developed two QSAR models based on machine learning for 3CLpro and RdRp of SARS-CoV-2 and validated these with high predictive performance [46]. Additionally, they identified some potential active inhibitors of 3CLpro and RdRp from FDA-approved drugs by using the models. In addition, Gawriljuk et al. implemented machine learning methods to develop predictive models from SARS-CoV-2 in vitro inhibition data. The best-performing model, with an AUC score of 0.81, was applied to predict the anti-SARS-CoV-2 activity of the compounds in FDA-approved drugs [49].

Many examples show that using machine learning models to screen active compounds can improve the success rate and save resources. In the present study, we implemented the random forest and support vector machine methods to establish the predictor using anti-SARS-CoV-2 bioactivity data from the ChEMBL database. The predictor can identify active compounds with anti-SARS-CoV-2 activity and novel structures from the TCMs in the Systems Pharmacology Database and Analysis Platform (TCMSP). Additionally, based on the predicted anti-SARS-CoV-2 compounds, we identified anti-SARS-CoV-2 medicinal plants through enrichment analysis. To the best of our knowledge, this is the first implementation of establishing an anti-SARS-CoV-2 compound predictor based on cell phenotype data to screen natural products from TCM.

## 2. Result

### 2.1. Development of anti-SARS-CoV-2 Compound Predictor

The development process of the anti-SARS-CoV-2 compound predictor is shown in Figure 1. The obtained dataset was highly imbalanced, containing 105 active anti-SARS-CoV-2 compounds and 3866 inactive anti-SARS-CoV-2 compounds. To solve the imbalanced classification problem, we used the undersampling method before establishing the classification models. Finally, a benchmark dataset including 105 active anti-SARS-CoV-2 compounds and 109 inactive anti-SARS-CoV-2 compounds was obtained (Appendix A). Then, the benchmark dataset was used to establish classification models using machine learning methods of RF and SVM. Using the parameter grid search and 10-fold cross-validation strategy, the optimal number of trees was chosen as 900 in the RF model, and the optimal C parameter was chosen as 50 in the SVM model (Appendix A). After training, parameter optimization, and model evaluation, the optimal SVM and RF models were established. Both models were used to predict the anti-SARS-CoV-2 activity of compounds from the TCMSP.

### 2.2. Performance of the RF and SVM Models

We used the FP2 molecular fingerprint of compounds as the input feature of RF and SVM models and established classification models after determining the best parameters. The overall performance of the RF and SVM models is quantified by several classification evaluation indicators including accuracy, precision, recall, F1 score, and AUC. We took the average values of the indicators from 10-fold cross-validation as the final results. The average accuracy, precision, recall, and F1 scores were 0.85, 0.86, 0.84, and 0.85 for the RF model and 0.84, 0.86, 0.81, and 0.83 for the SVM model, respectively (Figure 2a). The average AUC values for the RF and SVM models were 0.90 and 0.90 (Figure 2b), which reveals the high effectiveness of the models.

### 2.3. Accuracy of More Than 72% from the External Dataset

To evaluate the predictive generalizability of the models, we also collected 107 experimentally tested active anti-SARS-CoV-2 drugs (IC50 < 10 μmol/L) from the DrugRepV database as the external dataset [50]. The prediction probability of the RF and SVM models for each compound is shown in Figure 3a (Appendix A). If the prediction probability is above 0.5 for the drug in the RF model, it is considered the active drug; the RF model predicted 82 compounds with an accuracy of 76.6%. If the prediction probability is above 0 for the drug in the SVM model, it is considered the active drug; the SVM model predicted 78 compounds with an accuracy of 72.9% (Figure 3b). Molnupiravir and Paxlovid (nirmatrelvir, ritonavir), approved by the FDA for clinical treatment of COVID-19, are also accurately predicted by the RF and SVM models. This indicates that both models have good generalizability.

### 2.4. Identification of Novel Anti-SARS-CoV-2 Compounds by Screening the TCMSP

The screening of all compounds from the TCMSP using RF and SVM models was completed to discover novel anti- SARS-CoV-2 compounds. The prediction probability of each compound in the RF and SVM models is shown in Figure 4a (Appendix A). Consensus scoring can improve prediction reliability [51]. We set a consensus scoring rule for determining the active anti-SARS-CoV-2 compound: the prediction probability is above 0 in the SVM model and simultaneously above 0.5 in the RF model. Based on the screening criteria, 1013 compounds were predicted as active and 5158 compounds were predicted as inactive in the RF and SVM models (Figure 4b). MOL000098 and MOL002008, MOL004128, MOL008785, MOL008806, and MOL008821 have the same structure (TC = 1) as CHEMBL50 and CHEMBL507100 which are in the active training set, respectively. Those six compounds with anti-SARS-CoV-2 activity have been verified in the experiment, and the IC50 of these compounds is between 2.3 and 3.9 μmol/L (Table 1). Furthermore, we evaluated the applicability domain (AD) of the models based on the Euclidean distances among all the compounds of the training set (see Section 4). Based on the distance matrix of the training set, the D value of 11.614 and σ of 2.660 were obtained. Finally, according to Equation (6), the applicability domain threshold (ADT) of the models was 12.944. There are four compounds outside the AD of the models (Appendix A). We found that only two compounds (MOL003422, MOL005051) of the 1013 predicted active compounds were outside the AD and two compounds (MOL000017, MOL002324) of the 5513 inactive compounds were outside the AD. This indicates the relative reliability of our prediction results. For a more rigorous analysis, we removed the compounds outside the AD in the subsequent analysis.

For the 1011 active anti-SARS-CoV-2 compounds within the AD, we used ClassyFire to classify them [52]. We found that most of them belong to flavonoids (390 compounds) (Figure 4c, Appendix A). The 105 known active anti-SARS-CoV-2 compounds are also classified by the ClassyFire, and 18 compounds belong to Carboxylic acids and derivatives. The top 10 categories for each dataset were shown. There are only two overlap structure categories (Organooxygen compounds and Carboxylic acids and derivatives) in both datasets. Therefore, our model can not only predict the original structure categories of the active training set but also predict other different structure categories. The RNA-dependent RNA polymerase (RdRp) and 3C-like protease (3CLpro) play important roles in the viral life cycle and are considered the most promising targets for drug discovery against SARS-CoV-2 [53]. Many studies have conducted virtual screening for these two targets [54,55]. Therefore, we analyzed the probable targets of these predicted hits through molecular docking. The 3CLpr inhibitor PF-07321332 (−7.0 kcal/mol) and the RdRp inhibitor Remdesivir (−5.3 kcal/mol) were selected as positive controls for the RdRp and 3CLpro proteins, respectively. The docking results showed that 15 compounds specifically docked to the 3CLpro protein, 221 compounds specifically docked to the RdRp protein, and 278 compounds could dock to both the 3CLpro and RdRp proteins (Appendix A).

To understand the differences in the physicochemical properties of the compounds between the active and inactive anti-SARS-CoV-2 compounds, we compared the molecular weight (MW), logP, number of hydrogen-bond donors (HBD), number of hydrogen-bond acceptors (HBA), topological polar surface area (TPSA), and number of rotatable bonds (RBN) of both groups of compounds (Figure 4d). There were significant differences (*p*-value < 0.0001) in those physicochemical properties between both groups in the Wilcoxon test. Active compounds have higher median values in MW (368.4 vs. 212.4), HBD (3 vs. 1), HBA (7 vs. 2), TPSA (111.1 vs. 35.3), and RBN (4 vs. 3) than inactive compounds, while logP (2.3 vs. 2.8) is lower. From the distribution of these physicochemical properties, most of the active compounds fit Lipinski’s Rule of Five [56].

Molecular fingerprint similarity was calculated between the 1011 active anti-SARS-CoV-2 compounds and the 77 FDA-approved antiviral drugs. There were low overall similarities between the two groups. The active anti-SARS-CoV-2 compounds and the FDA-approved antiviral drugs showed a mean molecular fingerprint similarity of less than 0.25. Furthermore, the maximum similarity of the molecular fingerprint is mostly concentrated in the range of 0.2–0.3. Additionally, 24 compounds have high similarity (TC: 0.8–1) with the FDA-approved antiviral drugs (Figure 4e). This indicates that most of the active anti-SARS-CoV-2 compounds are structurally novel.

### 2.5. Identification of 74 Potential Anti-SARS-CoV-2 Medicinal Plants through Enrichment Analysis

We mapped 1011 active anti-SARS-CoV-2 compounds to the 74 medicinal plants (p.adj < 0.05) using enrichment analysis. It was found that the 74 medicinal plants are widely distributed in 68 genera and 43 families (Appendix A), and we have classified them according to the functional properties of these medicinal plants. Most of the 74 anti-SARS-CoV-2 medicinal plants belong to ‘Antipyretic Detoxicate Drugs’, accounting for 18.9% (Figure 5a). The top 20 enriched medicinal plants are shown in Figure 5b, and we used Cytoscape to build a network of the top 20 predicted anti-SARS-CoV-2 medicinal plants and active compounds (Figure 5c). Most active anti-SARS-CoV-2 compounds exist in licorice, Epimrdii Herba, and Eucommiae Cortex. Among the 1011 active anti-SARS-CoV-2 compounds, 689 compounds come from a unique medicinal plant (Figure 5d). This indicates that these compounds have rich structural diversity and complexity, providing a wider biologically relevant chemical space [57].

## 3. Discussion

The SARS-CoV-2 virus has severely affected human life as well as economies around the world. Although several therapeutic drugs and vaccines are now available, the virus still evolves in the form of different strains. Viral variants of concern may emerge with dangerous resistance to the present vaccines and drugs. Therefore, it is an urgent requirement to identify and develop new drugs against the virus infection. TCM provides huge resources for developing antiviral drugs. Additionally, the application of machine learning in drug development facilitates the discovery of anti-SARS-CoV-2 drugs from medicinal plants. In this study, we constructed a benchmark dataset including 105 active and 109 inactive anti-SARS-CoV-2 compounds. The RF and SVM models that we trained using the benchmark dataset have good prediction performance, with mean AUC values of 0.90 and 0.90, respectively. Moreover, both models achieved more than 72% accuracy in the external dataset. Additionally, we evaluated the AD of the models based on the Euclidean distance method and used the models to screen compounds within the AD. A total of 1011 compounds were predicted to possess anti-SARS-CoV-2 activity from the 6526 herb-derived compounds from TCMSP using the predictor. Then, we inferred 74 medicinal plants that were significantly enriched in predicted anti-SARS-CoV-2 compounds by enrichment analysis.

Most of the predicted anti-SARS-CoV-2 compounds belong to the flavonoids category, accounting for 38.6% (390). Flavonoids are secondary plant metabolites that have a variety of biological activities such as anti-bacterial [58], anti-cancer [59], anti-inflammatory [60] and immunomodulatory effects [61]. Furthermore, previous studies showed that flavonoids have a strong antiviral capacity [62,63,64]. MOL000098 (quercetin) and MOL002008 (myricetin) also belong to the flavonoid category, and they can inhibit the infection ability of SARS-CoV-2 by targeting its 3CLpro [65,66]. Other categories also have antiviral properties, such as isoflavonoids [67], coumarins and derivatives [68], and aporphines [69]. The anti-viral properties of these categories of compounds might be applicable in fighting the COVID-19 pandemic.

SARS-CoV-2 virus infections can cause a cytokine storm, which induces excessive inflammation and immune response in the lung and other tissues; thus, COVID-19 is a multi-phasic disease [70,71]. In the theory of TCM, COVID-19 belongs to the category of ‘dampness-toxicity plague’, which has pathogenetic characteristics described as ‘dampness, heat, toxin, stasis, deficiency, and closure’ [72]. At the beginning of the SARS-CoV-2 outbreak in early 2020, the clinical application of TCM for the treatment of viral infections achieved good results in China. For example, Qingfei Paidu decoction (QFPD) and the Lianhua Qingwen capsule/granule (LHQW) with multiple components of medicinal plants were recommended for the treatment of the disease by the People’s Republic of China and the National Administration of Traditional Chinese Medicine [72]. There are at least 116 TCM herbal formulas composed of different medicinal plants [73,74]. Licorice and Scutellariae Radix were the common ingredients of herbal formulas, and these two medicinal plants were also predicted as anti-SARS-CoV-2 medicinal plants in our build predictor. Some studies have reported the components of licorice and Scutellariae Radix. Licorice-saponin A3 and glycyrrhetinic acid isolated from licorice can suppress SARS-CoV-2 by targeting nsp7 and the S-RBD, respectively [36]. Baicalein, an extract of Scutellariae Radix, inhibits the replication of SARS-CoV-2 by targeting 3CLpro [75]. Among the 74 anti-SARS-CoV-2 medicinal plants, 14 of them belong to the category of ‘Antipyretic Detoxicate Drugs’, which can relieve the dampness-toxicity pathogenesis of COVID-19 patients based on the theory of TCM [72]. These identified plants are widely distributed in 68 genera and 43 families, which provides a broader scope and vision for the screening of lead compounds of anti- SARS-CoV-2 drugs. These newly discovered anti- SARS-CoV-2 plants are worthy of more attention and further study.

In this study, the benchmark dataset mainly collected the cell-based assays which cells infected with virus strains BavPat1, USA-WA1/2020 strain, βCoV/KOR/KCDC03/2020, WA-1 strain-BEI #NR-52281, isolate France/IDF0372/2020. All strains were derived from the Alpha variation strain before 2021. Therefore, these predicted molecules may deal with the Alpha variation strain of coronavirus. The applicability of the results for other SARS-CoV-2 A variations is needed for further investigation. Furthermore, we investigated the probable targets of these predicted hits through molecular docking (see details in Section 4). The docking results showed that 15 compounds specifically docked to the 3CLpro protein, 221 compounds specifically docked to the RdRp protein, and 278 compounds could dock to both the 3CLpro and RdRp proteins (Appendix A). Other compounds may target other viruses or host proteins.

With the accumulation of biological data and the increase in the variety and complexity of data types, machine learning plays an important role in the integration of these data. Compared with molecular docking of classical virtual screening, virtual screening based on machine learning does not require structural information of the target, can effectively search large quantities of compounds, and can overcome the shortage of traditional virtual screening in terms of time and consumption of computation resources [76]. However, there are still many difficulties in the discovery of anti-SARS-CoV-2 compounds in silico by machine learning. Firstly, the accuracy of the prediction model is influenced by many factors, such as the quality of the benchmark datasets, the characterization of compounds, and the optimized model parameters [77,78,79]. The imbalance of training data is the major limitation for developing an anti-SARS-CoV-2 compound predictor. In this case, the classifier usually tends to predict input samples as the majority class, resulting in poor predictive accuracy for the minority class [80]. In general, there are two solutions for the class imbalance problem, which are oversampling and undersampling methods. The oversampling methods are used to increase the samples of the minority class, which may result in overfitting in the process of model construction. While the undersampling methods are used to reduce the samples of the majority class, some useful data present in the majority class may be deleted in this way [81]. To overcome the limitations of undersampling methods [82], we used similarity clustering for the majority classes to select the representative compound in each cluster. The benchmark dataset processed by this method improved the performance of prediction models in our study. In a previous study, the best model constructed based on cell phenotype data of anti-SARS-CoV-2 compounds had an AUC of 0.81, and our model achieved better predictive performance (AUC = 0.9). In this study, we have used both the 10-fold cross-validation method and the external dataset to validate our prediction models.

The definition of the active or inactive anti-SARS-CoV-2 compounds in this study is based on the in vitro experimental data. However, SARS-CoV-2 is a newly emerged coronavirus, and the available bioactivity data for the virus are little. A small amount of training data may lead to overfitting of the model and affect the generalization ability of the model, leading us to be unable to identify more types of chemical space. Furthermore, the compounds in this study were characterized by molecular fingerprints, which cannot reflect the complete structural features of the given compounds and are not suitable for macromolecular compounds [83,84]. So, we will continue to collect more bioactivity compound data for further optimization of the models. Finally, and most importantly, the compounds predicted by the models also need to be validated in vitro and in vivo experiments.

In summary, the predicted compounds and medicinal plants from the TCMSP in our build predictor offer an attractive starting point and a broader scope to explore potentially novel anti-SARS-CoV-2 agents. This study accelerates the in-depth analysis for the discovery of anti-SARS-CoV-2 agents from TCM.

## 4. Material and methods

### 4.1. Construction of the Benchmark Dataset

The initial dataset of SARS-CoV-2 bioactivity was extracted from the ChEMBL database (https://www.ebi.ac.uk/chembl/, Release28, accessed on 10 Feb 2022) [85]. To maintain consistency in the dataset, we selected the same cell type (Vero C1008). In accordance with previous studies, compounds with IC50 or EC50 less than 10 μmol/L were considered active anti-SARS-CoV-2 compounds, and compounds with an IC50 or EC50 higher than 100 μmol/L or an inhibition rate in the range of 0–5% were considered as inactive anti-SARS-CoV-2 compounds [86,87,88]. A total of 105 active compounds and 3866 inactive compounds from the ChEMBL database were retained after removing duplicate compounds. For enhancing the classification performance of models, we solved the imbalanced classification problem by using DataWarrior (v05.02.01) for similarity clustering (cut off = 0.8, clusters = 105) of inactive set and selecting the center compound of each cluster [89]. However, four compounds (CHEMBL500576, CHEMBL221265, CHEMBL1325592, CHEMBL317840) could not be clustered in the DataWarrior. We also added four compounds as inactive compounds. Finally, we obtained a benchmark dataset including 105 active anti-SARS-CoV-2 compounds and 109 inactive anti-SARS-CoV-2 compounds. Pybel, a python wrapper of Openbabel [90], can generate molecular fingerprints and perform molecular physicochemical property analysis by calculating the SMILES string of compounds. Studies have shown that the classification model of compound activity based on the FP2 molecular fingerprint features shows good prediction accuracy [77]. Therefore, we use the FP2 molecular fingerprint as the input feature of machine learning methods. The features of each compound were presented by the 1024 bits of FP2 molecular fingerprint calculated by pybel (v2.4.1).

### 4.2. Machine Learning Analysis of the Benchmark Dataset

To choose the appropriate machine learning methods to construct the anti-SARS-CoV-2 compound prediction model, we evaluate the predictive performance of different classification methods recommended by Scikit-learn (v0.19.2) including LogisticRegression, DecisionTreeClassifier, Support Vector Machine, GaussianNB, KNeighborsClassifier, RandomForestClassifier, AdaBoostClassifier, and GradientBoostingClassifier. The benchmark dataset was split into the training set (accounting for 75%) and the test set (accounting for 25%), and then the optimal parameters of the algorithm were determined using 10-fold cross-validation and a grid search strategy. The optimal model was selected by comparing the mean AUC of 10-fold cross-validation under different parameters. The results show that the support vector machine (SVM) and random forest (RF) methods perform best on the benchmark dataset (Appendix A). At the same time, in previous research, the SVM and RF methods have the best performance in predicting the activity of compounds [77]. Therefore, the benchmark dataset based on the FP2 molecular fingerprints and the SVM and RF methods were selected in the subsequent analysis.

#### 4.2.1. Random Forest Model

The random forest model (RF) for anti-SARS-CoV-2 compound prediction was established using the ‘ensemble’ module in Scikit-learn. The best number of trees (parameter n_estimators) was determined using a parameter grid search strategy, and other parameters were set as default values. The benchmark dataset was randomly divided into a training set (75%) and a test set (25%) using the ‘train_test_split’ function in Scikit-learn. The 10-fold cross-validation was applied to evaluate the generalization performance of the model with specified parameters. Then, we calculated the area under the curve (AUC) to assess the performance of models in the test set. The mean AUC was calculated after repeating the cross-validation for each given parameter. By comparing the highest mean AUC of 10-fold cross-validation under different parameters, the optimal model was selected.

#### 4.2.2. Support Vector Machine Model

The support vector machine (SVM) for anti-SARS-CoV-2 compound prediction was established using the ‘svm’ module in Scikit-learn. The radial basis function (rbf) was chosen as the kernel function to find a non-linear classifier. The optimal regularization parameter C was determined using a parameter grid search strategy, and other parameters were set as default values. The process of cross-validation and parameter optimization is described in the section on the random forest model. The optimal model was also selected based on the highest mean AUC of 10-fold cross-validation.

#### 4.2.3. Performance Evaluation

The performance of the optimal RF and SVM models was measured by the indicators of *accuracy*, *precision*, *recall*, and *F*1 score, which are calculated as follows:(1)$$accurary=\frac{TP+TN}{TP+TN+FP+FN}$$
(2)$$precision=\frac{TP}{TP+FP}$$
(3)$$recall=\frac{TP}{TP+FN}$$
(4)$$F1=\frac{2\times precision\times recall}{precision+recall}$$

True Positive (*TP*): the number of correctly predicted active anti-SARS-CoV-2 compounds; True Negative (*TN*): the number of correctly predicted inactive anti-SARS-CoV-2 compounds; False Positive (*FP*): the number of inactive anti-SARS-CoV-2 compounds predicted as active anti-SARS-CoV-2 compounds; False Negative (*FN*): the number of active anti-SARS-CoV-2 compounds predicted as inactive anti-SARS-CoV-2 compounds.

We also constructed an independent dataset to evaluate the generalizability of the models from the DrugRepV database [50], which contained an experimentally tested active drug (IC50 < 10 μmol/L) against SARS-CoV-2. A total of 107 active anti-SARS-CoV-2 drugs were selected after removing the compounds that are also present in the training set.

### 4.3. Anti-SARS-CoV-2 Compound Prediction

The benchmark dataset with optimal parameters was used to establish the final RF and SVM models. We collected all herb-derived compounds (6526) from the Traditional Chinese Medicines for Systems Pharmacology Database and Analysis Platform (TCMSP, https://old.tcmsp-e.com/, accessed on 25 May 2022) to predict anti-SARS-CoV-2 activity using the RF and SVM model. Consensus scoring can improve prediction reliability, so the compound that predicted anti-SARS-CoV-2 activity in both RF and SVM models was designated as the active anti-SARS-CoV-2 compound. Moreover, we collected 77 FDA-approved antiviral drugs to measure their similarity with active compounds [91,92]. The similarity between compounds is measured by Tanimoto Coefficient (*TC*) as follows [93]:(5)$$TC(A,B)=\frac{c}{a+b-c}$$
where *a* means the number of molecular fingerprint features of ‘1’ in compound A; *b* means the number of molecular fingerprint features of ‘1’ in compound B; *c* means the shared number of molecular fingerprint features of ‘1’ between compound A and compound B.

### 4.4. Analysis of Applicability Domain (AD) of Models

The applicability domain (AD) is a theoretical region in the chemical space surrounding both the model descriptors and modeled response. Due to the limitation of the chemical space of the training sets, it is impractical to predict a whole universe of chemicals using a single QSAR model [94]. Therefore, AD should be seriously considered in any QSAR-based predictive model. The setting of AD can prevent excessive prediction bias from the large characteristic difference between the query compound and the training set. It means that the model prediction can be considered relatively reliable if the query compound is distributed in AD; otherwise, it is less reliable. There are various methods for calculating the AD of QSAR models, such as distance-based methods, and probability density distribution [95]. In this study, we employed the Euclidean distance method, which has been widely used in previous studies [96,97,98]. The distribution of Euclidean distances among all the compounds in the training set was computed to produce the applicability domain threshold (*ADT*) [99]. Then, we computed the distance of the query compound to its nearest neighbor in the training set. If the distance is beyond the *ADT*, the prediction is considered unreliable. The *ADT* was calculated as follows in Equation (6).
(6)ADT=D+Zσ
where *Z* is an empirical similarity threshold parameter (default is 0.5); the *D* and *σ* are the average and standard deviation of all the Euclidian distances in the multidimensional descriptor space among all the compounds in the training set, respectively.

### 4.5. Molecular Docking

We further investigated the probable targets of these predicted hits using molecular docking. The RNA-dependent RNA polymerase (RdRp) and 3C-like protease (3CLpro) encoded by the SARS-CoV-2 genome play important roles in the viral life cycle and are considered the most promising targets for drug discovery against SARS-CoV-2 [53]. The crystal 3D structures of 3CLpro (PDB ID: 7VH8) and RdRp (PDB ID: 7BV2) were downloaded from the Protein Data Bank. Water molecules, inhibitors, and unrelated chemical complexes were removed from 7VH8 and 7BV2 using Discovery Studio Visualizer (v2019). For each complex, the center of the search box was set as the center of the geometry of the crystallized ligand. Through Discovery Studio Visualizer, we obtained the docking centers of RdRp (center: X = 91.51, Y = 92.38, Z = 103.73) and 3CLpro (center: X = −18.76, Y = 17.14, Z = −25.14). The 1013 predicted active compounds from the TCMSP were used for molecular docking. Then, the receptors and ligands were converted to PDBQT format using MGLtools (prepare_ligand4.py and prepare_receptor4.py, v1.5.7). Finally, docking analysis was performed using the AutoDock Vina (v1.1.2) software. The box size was set to 15 *15 *15 Å, and other parameters were set to default. After docking, the 3CLpr inhibitor PF-07321332 (-7.0 kcal/mol) and the RdRp inhibitor Remdesivir (−5.3 kcal/mol) were selected as positive controls for the RdRp and 3CLpro proteins, respectively.

### 4.6. Inferring the Anti-SARS-CoV-2 Medicinal Plants by Enrichment Analysis

We collected all TCMs (499) and herb-derived compounds (6526) from the TCMSP database as the reference background set. Based on the active compounds predicted from the RF and SVM models, we used the enricher function of ‘clusterProfiler’ in the R library to perform enrichment analysis [100]. Only medicinal plants with p.adj < 0.05 were considered potential anti-SARS-CoV-2 medicinal plants. Then, a network consisting of active compounds and medicinal plants was established and visualized with Cytoscape (v3.8.2).

### 4.7. Molecular Descriptors

Based on the SMILES string of compounds, we calculated molecular descriptors using the rdkit (2020.09.1.0) of the Python library, including the molecular weight (MW), logP, number of hydrogen-bond donors (HBD), number of hydrogen-bond acceptors (HBA), topological polar surface area (TPSA), and number of rotatable bonds (RBN).

## Figures and Tables

**Figure 1 molecules-28-00208-f001:**
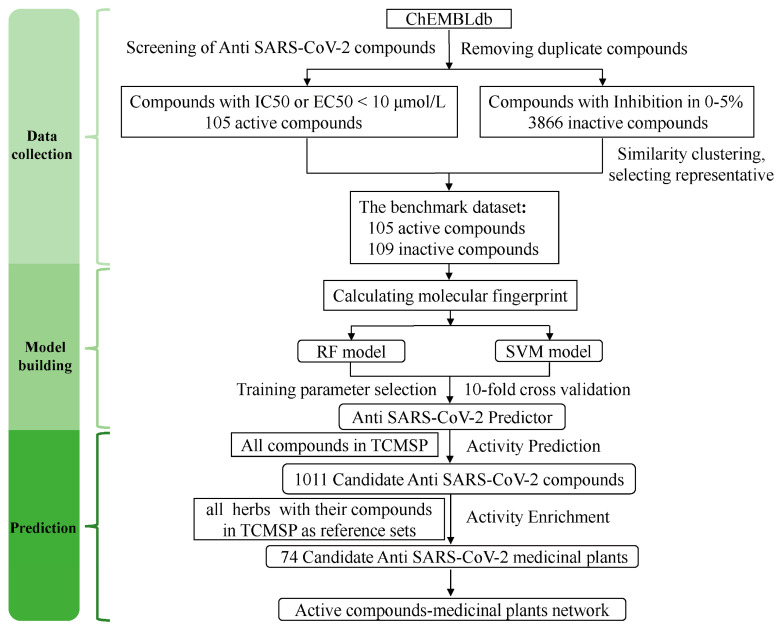
The workflow of the construction of the anti-SARS-CoV-2 compound predictor. The benchmark dataset contained the active and inactive anti-SARS-CoV-2 compounds obtained from the ChEMBL database. We constructed the anti-SARS-CoV-2 compound predictor to predict the anti-SARS-CoV-2 activity of herbs-derived compounds from the TCMSP by the RF and SVM models. Then, we mapped the predicted active compounds to identify anti-SARS-CoV-2 medicinal plants using enrichment analysis.

**Figure 2 molecules-28-00208-f002:**
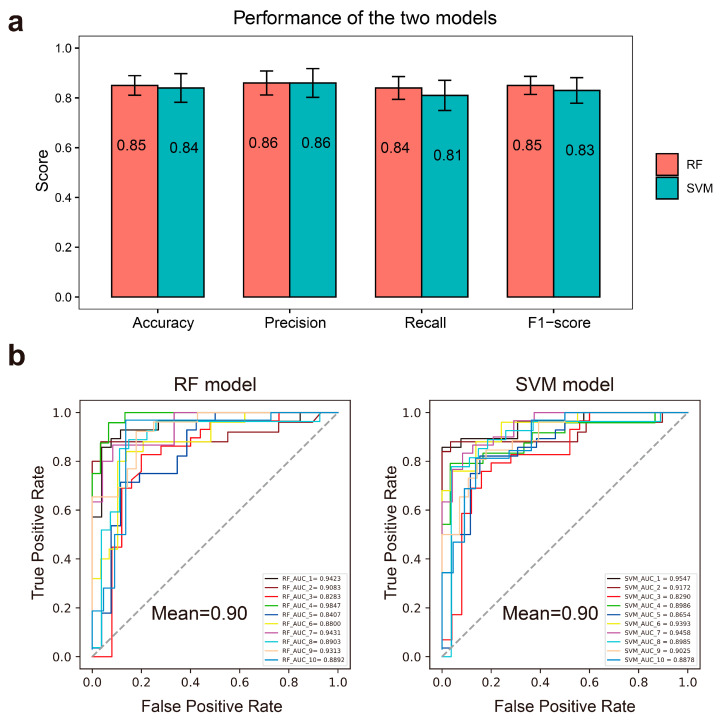
The performance of RF and SVM models. (**a**). Bar plots show the mean values of 10-fold cross-validation derived from the RF model (red) and SVM model (green), respectively. The error bars represent the standard deviations. (**b**). The receiver operating characteristic (ROC) curve of the RF (left) and SVM (right) models in the test dataset of each cross-validation.

**Figure 3 molecules-28-00208-f003:**
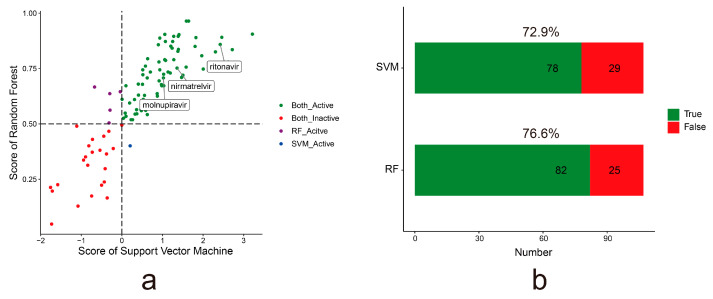
The predictive results of the anti-SARS-CoV-2 predictor in the external dataset. (**a**). The prediction probability of RF and SVM models for each compound from the external dataset. Each dot represents a compound. The green and red dots denote the active and inactive anti-SARS-CoV-2 compound supported by both the RF and SVM models, respectively. The purple dot denotes the active anti-SARS-CoV-2 compound supported by only the RF model. The blue dot denotes the active anti-SARS-CoV-2 compound supported by only the SVM model. (**b**). The bar plots show the accuracy of the RF and SVM models in the external dataset.

**Figure 4 molecules-28-00208-f004:**
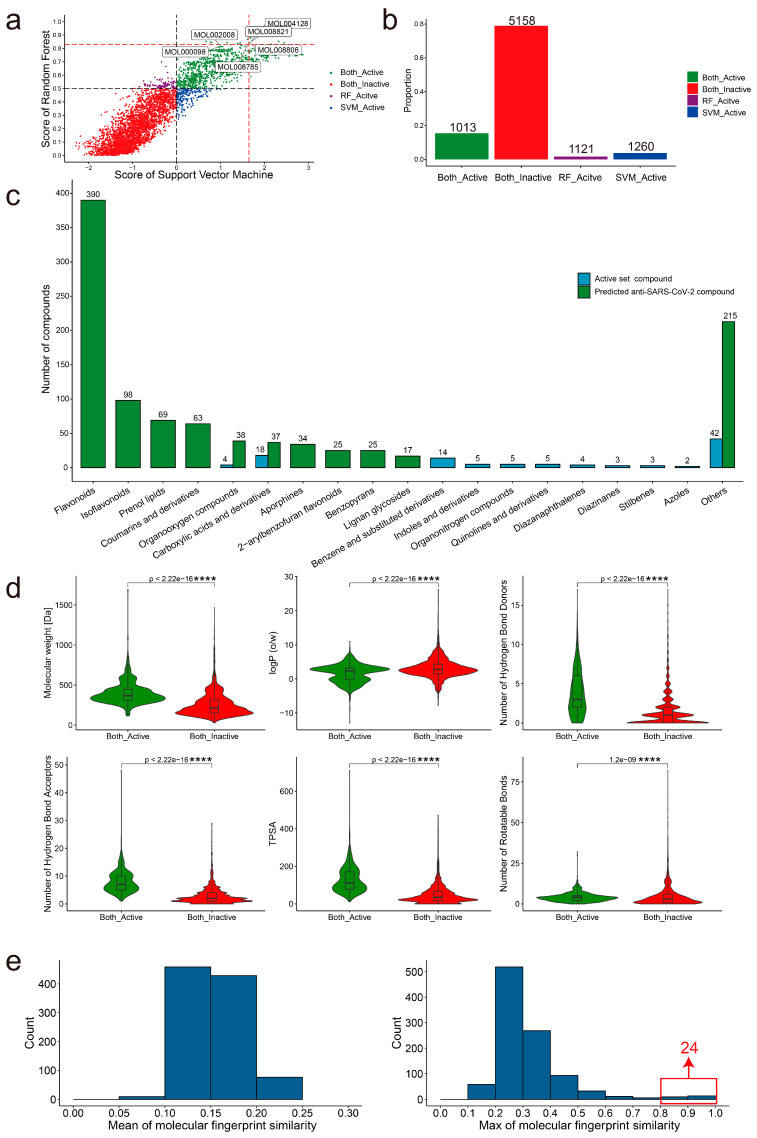
The predictive results of anti-SARS-CoV-2 predictor in the TCMSP and the structural characteristics of the active and inactive compounds. (**a**). The prediction probability of the RF and SVM models for each compound from the TCMSP. Each dot also represents a compound. (**b**). The bar plot shows the number of compound labels predicted by the two models. The green and red bars mean the number of active and inactive anti-SARS-CoV-2 compounds supported by both the RF and SVM models, respectively. The purple bar means the number of active anti-SARS-CoV-2 compounds supported by the RF model. The blue bar means the number of active anti-SARS-CoV-2 compounds supported by the SVM model. (**c**). The top 10 categories of the 1011 active anti-SARS-CoV-2 compounds and 105 active compounds of training set, respectively. (**d**). The violin plot displays the comparisons of the MW, logP, HBD, HBA, TPSA, and RBN between active and inactive compounds. “****” means the significance levels (*p* < 0.0001). (**e**). The histogram plots show the average (left) and maximum (right) similarity of the molecular fingerprint of 1011 active anti-SARS-CoV-2 compounds and 77 FDA-approved antiviral drugs.

**Figure 5 molecules-28-00208-f005:**
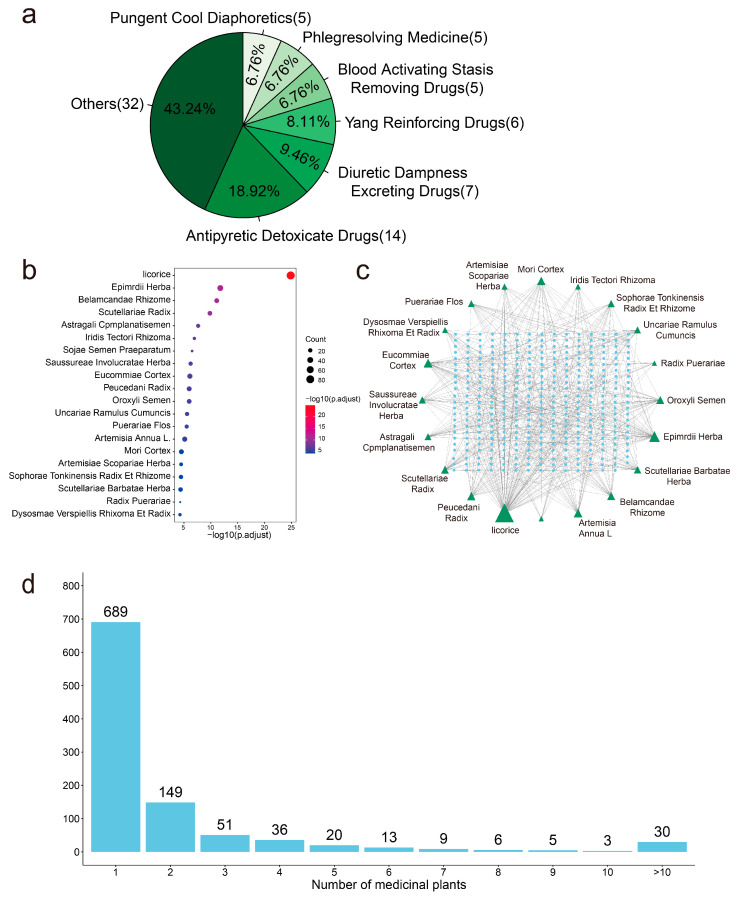
The predicted anti-SARS-CoV-2 medicinal plants. (**a**). The classification of functional characteristics of 74 anti-SARS-CoV-2 medicinal plants. (**b**). Top 20 medicinal plants enriched from 1011 active anti-SARS-CoV-2 compounds by enrichment analysis. (**c**). The interaction network of top 20 predicted anti-SARS-CoV-2 medicinal plants and active compounds. (**d**). The bar plot shows the number of medicinal plant sources of the compounds.

**Table 1 molecules-28-00208-t001:** Six compounds have the same structure as the active training set (TC = 1).

Qurey Compound ^1^	Match Compound ^2^	Max TC ^3^	IC50 ^4^ (μmol/L)	SVM Probability	SVM Label	RF Probability	RF Label
MOL000098	CHEMBL50	1	2.3	1.00	1	0.82	1
MOL002008	CHEMBL50	1	2.3	1.00	1	0.82	1
MOL004128	CHEMBL507100	1	3.9	1.65	1	0.84	1
MOL008785	CHEMBL507100	1	3.9	1.65	1	0.84	1
MOL008806	CHEMBL507100	1	3.9	1.65	1	0.84	1
MOL008821	CHEMBL507100	1	3.9	1.65	1	0.84	1

^1^ Compounds of TCMSP; ^2^ The compound in the active training set has the highest similarity with query compound; ^3^ The similarity value of molecular fingerprint between the query compound and the match compound. ^4^ IC50 of the second column compound in Vero C1008 cells.

## Data Availability

Data provided in Appendix A. The code and associated dataset were hosted on Github (https://github.com/daishaoxing/anti-SARS-CoV-2, accessed on 28 October 2022).

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
