# Peer review of "In Silico Identification of Anti-SARS-CoV-2 Medicinal Plants Using Cheminformatics and Machine Learning"

_molecules, 2022, doi:10.3390/molecules28010208_

Round 1
Reviewer 1 Report
In this manuscript, authors used machine learning-based virtual screening to identify potential anti-SARS-CoV-2 molecules from TCM. This manuscript is well-written and easy to understand. I have the following questions for the authors -
Among different machine learning methods available why only random forest and support vector machines are used?
From 3866 inactive compounds, authors received 105 clusters (line 282) then how were 109 inactive compounds selected in the inactive dataset?
How these predicted molecules would deal with different variations of coronavirus that are present and may arise in the future as there is no specific target known for these molecules?
It would be of much interest if authors could investigate the probable targets of these predicted hits.
Authors should provide the code and associated dataset for reproducibility.
Author Response
Reviewer 1 comments and response
Comment 1: In this manuscript, authors used machine learning-based virtual screening to identify potential anti-SARS-CoV-2 molecules from TCM. This manuscript is well-written and easy to understand. I have the following questions for the authors.
Among different machine learning methods available why only random forest and support vector machines are used?
Response: Thanks for your professional review work on our article. As your comment, there are many different classification methods recommended by Scikit-learn(v0.19.2) including LogisticRegression, DecisionTreeClassifier, Support Vector Machine (SVM), GaussianNB, KNeighborsClassifier, RandomForestClassifier (RF), AdaBoostClassifier, and GradientBoostingClassifier. Firstly, based on our previous analysis, the SVM and RF methods have the best performance in predicting the activity of compounds [67]. To further determine the appropriate machine learning methods for the construction of the anti-SARS-CoV-2 prediction model, we evaluate the predictive performance of different classification methods. The benchmark dataset was split into the training set (75%) and the test set (25%), and then the optimal parameters of the algorithm are determined using 10-fold cross validation and a grid search strategy. The optimal model was selected by comparing the mean AUC of 10-fold cross validation under different parameter. The results show that the support vector machine (SVM) and random forest (RF) methods perform best on the benchmark dataset (Figure 1). Therefore, the benchmark dataset based on FP2 molecular fingerprints, and the SVM and RF methods were used in this study. We have added these contents in the revised manuscript (Line 368-381, Page 13) and in the Supplementary Materials (Supplementary Figure 2).
Comment 2: From 3866 inactive compounds, authors received 105 clusters (line 282) then how were 109 inactive compounds selected in the inactive dataset?
Response: Thanks for your careful checks. We feel sorry for not explaining it clearly. For enhancing classification performance of models, we solved the imbalanced classification problem by similarity clustering of inactive set using DataWarrior (v05.02.01). We got 105 clusters (cut off =0.8, clusters = 105) and selected the center compound of each cluster. However, there are four compounds (CHEMBL500576, CHEMBL221265, CHEMBL1325592, CHEMBL317840) could not be clustered in the DataWarrior. We also added four compounds as inactive compounds. Finally, we got a benchmark dataset including 105 active anti-SARS-CoV-2 compounds and 109 inactive anti-SARS-CoV-2
compounds. We have added this description in the manuscript (Line 353-366, Page 12-13).
Supplementary Figure 2. The AUC value of different machine learning classification methods for benchmark datasets. (a). Bar plots show the mean AUC values of 10-fold cross validation derived from different machine learning classification methods for benchmark datasets. The error bars represent the standard deviations. (b). Optimal parameters and mean AUC value of different machine learning classification methods for benchmark datasets.
Comment 3: How these predicted molecules would deal with different variations of coronavirus that are present and may arise in the future as there is no specific target known for these molecules?
It would be of much interest if authors could investigate the probable targets of these predicted hits.
Response: Thank you for your precious comments and advice. 1) In this study, the benchmark dataset mainly collected the cell-based assays which cells infected with virus strains BavPat1, USA-WA1/2020 strain, βCoV/KOR/KCDC03/2020, WA-1 strain-BEI #NR-52281, isolate France/IDF0372/2020. All strains were derived from Alpha variation strain before 2021. Therefore, these predicted molecules may deal with Alpha variation strain of coronavirus. The applicability of the results for other SARS-CoV-2 A variations is needed further investigation. 2) As your comment, we investigated the probable targets of these predicted hits through molecular docking. The RNA-dependent RNA polymerase (RdRp) and 3C-like protease (3CLpro) encode by SARS-CoV-2 genome play important roles in the viral life cycle and are considered the most promising targets for drug discovery against SARS-CoV-2. We used AutoDock Vina (v1.1.2) to conduct virtual screening of 1013 active anti-SARS-CoV-2 compounds against RdRp and 3CLpro proteins, respectively. The 3D structures of the RdRp (PDB ID: 7BV2) and 3CLpro (PDB ID: 7VH8) were downloaded from the Protein Data Bank. The ligand center of the complex is used as the docking center. 3CLpr inhibitor PF-07321332 (-7.0 kcal/mol) and RdRp inhibitor Remdesivir (-5.3 kcal/mol) were selected as positive controls for RdRp and 3CLpro proteins, respectively. The docking results showed that 15 compounds specifically docked to the 3CLpro protein, 223 compounds specifically docked to the RdRp protein, and 278 compounds could dock to both the 3CLpro and RdRp proteins. Other compounds may target other viruses or host proteins (Line 294-304, Page 11, Supplementary Figure 3).
Supplementary Figure 3. The pie chart showing the number of compounds against RdRp, 3CLpro and other proteins based on the docking results.
Comment 4: Authors should provide the code and associated dataset for reproducibility.
Response: Thank you for your advice. As your comment, the code and associated dataset has been hosted on Github (https://github.com/daishaoxing/anti-SARS-CoV-2) (Line 464-465, Page 15).
In summary, we thank the reviewer’s comments again. We have integrated the above contents in the revised manuscript and performed language-editing using a paid editing service. We hope the new version will meet your standards and requirements.

Reviewer 2 Report
This paper reported the predictor established with the methods of the random forest (RF) and support vector machine (SVM), and the resulting identification of active compounds against SARS-CoV-2 from Traditional Chinese Medicines in Systems Pharmacology Database and Analysis Platform (TCMSP). Of note, the identified compounds covered the reported active compounds, showing the good prediction accuracy, and showed the most of dissimilarity with the FDA-approved antiviral drugs. The topic fits the scope of the journal and may accelerate the discovery and development of anti-SARS-CoV-2 agents. In general, the manuscript is well-organized and the references as well as experiments can support the conclusions. Even though, the necessary experiments need to be performed and the key issues are required to be addressed before its publication on Molecules.
1. In the introduction part, the previously reported screening campaigns of Traditional Chinese Medicines as well as the performance/results are required to be introduced.
2. The prediction/identification method in this study is mostly based on the structure similarity and the included structure categories in the training set are required to be introduced.
3. The comparison of the prediction method in this study with the classical virtual screening is required to be discussed.
4. The limitation of this prediction method is required to be discussed.
Author Response
Reviewer 2 comments and response
Comment 1: This paper reported the predictor established with the methods of the random forest (RF) and support vector machine (SVM), and the resulting identification of active compounds against SARS-CoV-2 from Traditional Chinese Medicines in Systems Pharmacology Database and Analysis Platform (TCMSP). Of note, the identified compounds covered the reported active compounds, showing the good prediction accuracy, and showed the most of dissimilarity with the FDA-approved antiviral drugs. The topic fits the scope of the journal and may accelerate the discovery and development of anti-SARS-CoV-2 agents. In general, the manuscript is well-organized and the references as well as experiments can support the conclusions. Even though, the necessary experiments need to be performed and the key issues are required to be addressed before its publication on Molecules.
Response: Thank you for your affirmation and comments. Our study aims to discover potential anti-SARS-CoV-2 compound from TCM through machine learning and provided several medicinal plants with potential anti-SARS-CoV-2 activity, which offer an attractive starting point and a broader scope to mine for potentially novel an-ti-SARS-CoV-2 drugs. As your comments, the necessary experiments need to be performed. In this study, we have used 10-fold cross validation and the external dataset to validate our results and prediction models. We will create conditions to perform experiments for further validation in future. We have discussed this limitation in the revised manuscript (Lines 294-339, page 11-12).
Comment 2: In the introduction part, the previously reported screening campaigns of Traditional Chinese Medicines as well as the performance/results are required to be introduced.
Response: Thank you for your valuable comments. We have checked the literatures carefully and added more references on screening campaigns of Traditional Chinese Medicines into the introduction part in the revised manuscript (Lines 87-99, page 2).
Comment 3: The prediction/identification method in this study is mostly based on the structure similarity and the included structure categories in the training set are required to be introduced.
Response:We are grateful for the suggestion. We have added a more detailed analysis of structure categories of active training set. For the 1013 predicted anti-SARS-CoV-2 compounds, we used the method ClassyFire to classify them (Figure 4c). We found that most of them belong to flavonoids (390 compounds). The 105 known active anti-SARS-CoV-2 compounds also classified by the ClassyFire, and 18 compounds belong to Carboxylic acids and derivatives. The top 10 categories for each dataset were shown. There are only two overlap structure categories (Organooxygen compounds and Carboxylic acids and derivatives) in both datasets. Therefore, our model can not only predict the original structure categories of the active training set, but also predict other different structure categories. Previous studies also reported structural diverse compounds by using machine learning. We added this content in the revised manuscript (Lines 180-188, page 6, Figure 4c).
Figure 4C. The bar chart showing the number of compounds in each structure category classified by the method ClassyFire.
Comment 4: The comparison of the prediction method in this study with the classical virtual screening is required to be discussed.
Response: Thank you for the suggestion. We have introduced the studies about screening of anti-SARS-CoV-2 compounds using machine learning or molecular docking methods (Lines 74-100, page 2). We also discussed the comparison between your method and the other methods (Lines 305-310, page 12).
Comment 5: The limitation of this prediction method is required to be discussed.
Response: We deeply appreciate the reviewer’s suggestion. We have discussed the limitations of this study in the revised manuscript (Lines 294-339, page 11-12).
In summary, we thank the reviewer’s comments again. We have integrated the above contents in the revised manuscript and performed language-editing using a paid editing service. We hope the new version will meet your standards and requirements.

Reviewer 3 Report
The authors proposed in silico studies to identify the anti-SARS-CoV-2 medicinal plants using chemoinformatics and machine learning approaches. However, they did not consider already published studies on the very same topic to discuss their results. Also, they did not validate their results experimentally. Hundreds of theoretical studies have been published on this topic, and at this stage, without experimental validations, this manuscript's results will have almost no significant impact. Therefore, I recommend the rejection of this article in its current form.
Author Response
Reviewer 3 comments and response
The authors proposed in silico studies to identify the anti-SARS-CoV-2 medicinal plants using chemoinformatics and machine learning approaches. However, they did not consider already published studies on the very same topic to discuss their results. Also, they did not validate their results experimentally. Hundreds of theoretical studies have been published on this topic, and at this stage, without experimental validations, this manuscript's results will have almost no significant impact. Therefore, I recommend the rejection of this article in its current form.
Response: Thanks for your review work on our article. Based on your comments, we have made extensive modification in the previous version as follows.
1) We have introduced the studies about screening of anti-SARS-CoV-2 compounds using machine learning or molecular docking methods (Lines 74-100, page 2). We also discussed the comparison between your method and the other methods (Lines 305-310, page 12).
2). We have introduced the literatures about the anti-virus studies of Chinese herbal medicines (Lines 49-73, page 2).
3) We have compared the predictive performance of many different machine learning methods recommended by Scikit-learn(v0.19.2) including LogisticRegression, DecisionTreeClassifier, Support Vector Machine (SVM), GaussianNB, KNeighborsClassifier, RandomForestClassifier (RF), AdaBoostClassifier,and GradientBoostingClassifier. The benchmark dataset was split into the training set (75%) and the test set (25%), and then the optimal parameters of the algorithm are determined using 10-fold cross validation and a grid search strategy. The optimal model was selected by comparing the mean AUC of 10-fold cross validation under different parameter. The results show that the support vector machine (SVM) and random forest (RF) methods perform best on the benchmark dataset. Therefore, the benchmark dataset based on FP2 molecular fingerprints, and the SVM and RF methods were used in this study. We have added these contents in the revised manuscript (Line 368-381, Page 13) and in the Supplementary Materials (Supplementary Figure 2).
4) We have investigated the probable targets of these predicted hits through molecular docking. The RNA-dependent RNA polymerase (RdRp) and 3C-like protease (3CLpro) encode by SARS-CoV-2 genome play important roles in the viral life cycle and are considered the most promising targets for drug discovery against SARS-CoV-2. We used AutoDock Vina (v1.1.2) to conduct virtual screening of 1013 active anti-SARS-CoV-2 compounds against RdRp and 3CLpro proteins, respectively. The 3D structures of the RdRp (PDB ID: 7BV2) and 3CLpro (PDB ID: 7VH8) were downloaded from the Protein Data Bank. The ligand center of the complex is used as the docking center. 3CLpr inhibitor PF-07321332 (-7.0 kcal/mol) and RdRp inhibitor Remdesivir (-5.3 kcal/mol) were selected as positive controls for RdRp and 3CLpro proteins, respectively. The docking results showed that 15 compounds specifically docked to the 3CLpro protein, 223 compounds specifically docked to the RdRp protein, and 278 compounds could dock to both the 3CLpro and RdRp proteins. Other compounds may target other viruses or host proteins (Line 294-304, Page 11, Supplementary Figure 3).
5) The code and associated dataset have been hosted on Github (https://github.com/daishaoxing/anti-SARS-CoV-2) (Line 464-465, Page 15).
6) We have added a more detailed analysis of structure categories of active training set. For the 1013 predicted anti-SARS-CoV-2 compounds, we used the method ClassyFire to classify them (Figure 4c). We found that most of them belong to flavonoids (390 compounds). The 105 known active anti-SARS-CoV-2 compounds also classified by the ClassyFire, and 18 compounds belong to Carboxylic acids and derivatives. The top 10 categories for each dataset were shown. There are only two overlap structure categories (Organooxygen compounds and Carboxylic acids and derivatives) in both datasets. Therefore, our model can not only predict the original structure categories of the active training set, but also predict other different structure categories. We added this content in the revised manuscript (Lines 180-188, page 6, Figure 4c).
7). Currently, virtual screening of anti-SARS-CoV-2 compounds mainly focus on the FDA approved drugs and synthetic compounds. The natural compounds from medical plants were seriously neglected. Furthermore, manly screening experiments are target-based, which target Mpro and 3CLpro proteins. Our study is based on cell phenotype data and use machine learning (RF and SVM) to screen potential natural compounds against SARS-CoV-2. The performance of our method is better than previous similar method (Lines 305-310, page 12).
8) Our study aims to discover potential anti-SARS-CoV-2 compound from TCM through machine learning and provided several medicinal plants with potential anti-SARS-CoV-2 activity, which offer an attractive starting point and a broader scope to mine for potentially novel an-ti-SARS-CoV-2 drugs. As your comments, the necessary experiments need to be performed. In this study, we have used both the 10-fold cross validation method and the external dataset to validate our prediction models. We will create conditions to perform experiments for further validation in future (Lines 294-339, page 11-12).
In summary, we thank the reviewer’s comments again. We have integrated the above contents in the revised manuscript and performed language-editing using a paid editing service. We hope the new version will meet your standards and requirements.
Supplementary Figure 2. The AUC value of different machine learning classification methods for benchmark datasets. (a). Bar plots show the mean AUC values of 10-fold cross validation derived from different machine learning classification methods for benchmark datasets. The error bars represent the standard deviations. (b). Optimal parameters and mean AUC value of different machine learning classification methods for benchmark datasets.
Supplementary Figure 3. The pie chart showing the number of compounds against RdRp, 3CLpro and other proteins based on the docking results.
Figure 4C. The bar chart showing the number of compounds in each structure category classified by the method ClassyFire.

Reviewer 4 Report
The paper is devoted to a very urgent problem - the search for new effective antiviral agents against the SARS-CoV-2 virus. The authors quite rationally applied the QSAR approach for the subsequent screening of Traditional Chinese Medicines in Systems Pharmacology Database. The paper describes in sufficient detail the methods of research and the results obtained.
The reviewer has one, but a very serious remark. Unfortunately, the manuscript does not contain any information about the applicability domain (AD) of the used QSAR models. It is not known whether the AD was assessed, if so, by what method, whether all the studied molecules corresponded to AD. This is especially important to know, since the studied compounds are very structurally diverse. The reliability of the obtained results depends on the clarification of this problem.
Author Response
Reviewer 4 comments and response
The paper is devoted to a very urgent problem - the search for new effective antiviral agents against the SARS-CoV-2 virus. The authors quite rationally applied the QSAR approach for the subsequent screening of Traditional Chinese Medicines in Systems Pharmacology Database. The paper describes in sufficient detail the methods of research and the results obtained.
The reviewer has one, but a very serious remark. Unfortunately, the manuscript does not contain any information about the applicability domain (AD) of the used QSAR models. It is not known whether the AD was assessed, if so, by what method, whether all the studied molecules corresponded to AD. This is especially important to know, since the studied compounds are very structurally diverse. The reliability of the obtained results depends on the clarification of this problem.
Response: We deeply appreciate your comments.
1) We have added the applicability domain of the QSAR models in the revised manuscript (Lines 87-99, page 2). Machine learning methods are widely applied in almost all stages of the discovery and development of new drugs [37]. Moreover, virtual screening using machine learning methods is far more efficient than traditional drug discovery processes [38]. Some studies have adopted machine learning or deep learning methods to deal with the COVID-19 pandemic, especially aimed at repurposing existing drugs to treat the disease. They will be divided into two main groups: target-based and phenotype-based drug discovery methods [39-43]. For example, Ivanov et al. developed two QSAR models based on machine learning for 3CLpro and RdRp of SARS-Cov-2 and validated these with high predictive performance [40]. Additionally, they identified some potential active inhibitors of 3CLpro and RdRp from FDA-approved drugs by using the models. In addition, Gawriljuk et al. implemented machine learning methods to develop predictive models from SARS-CoV-2 in vitro inhibition data. The best performing model, with an AUC score of 0.81, was applied to predict the anti SARS-Cov-2 activity of the compounds in FDA-approved drugs. Our study aims to discover potential anti-SARS-CoV-2 compound from TCM through machine learning and provided several medicinal plants with potential anti-SARS-CoV-2 activity, which offer an attractive starting point and a broader scope to mine for potentially novel an-ti-SARS-CoV-2 drugs.
2) As your comments, the necessary experiments need to be performed. In this study, we have used both the 10-fold cross validation method and the external dataset to validate our prediction models. We will create conditions to perform experiments for further validation in future. We have discussed the limitations of this study in the revised manuscript (Lines 294-339, page 11-12).
3) Indeed, our predicted compounds are very structurally diverse. We have added a more detailed analysis of structure categories of active training set. For the 1013 predicted anti-SARS-CoV-2 compounds, we used the method ClassyFire to classify them (Figure 4c). We found that most of them belong to flavonoids (390 compounds). The 105 known active anti-SARS-CoV-2 compounds also classified by the ClassyFire, and 18 compounds belong to Carboxylic acids and derivatives. The top 10 categories for each dataset were shown. There are only two overlap structure categories (Organooxygen compounds and Carboxylic acids and derivatives) in both datasets. Therefore, our model can not only predict the original structure categories of the active training set, but also predict other different structure categories. Previous studies also reported structural diverse compounds by using machine learning. We added this content in the revised manuscript (Lines 180-188, page 6, Figure 4c).
4) In addition to the above three points, we have made extensive modification in the previous version as follows.
- A) We have introduced the studies about screening of anti-SARS-CoV-2 compounds using machine learning or molecular docking methods (Lines 74-100, page 2). We also discussed the comparison between your method and the other methods (Lines 305-310, page 12).
- B) We have introduced the literatures about the anti-virus studies of Chinese herbal medicines (Lines 49-73, page 2).
- C) We have compared the predictive performance of many different machine learning methods recommended by Scikit-learn(v0.19.2) including LogisticRegression, DecisionTreeClassifier, Support Vector Machine (SVM), GaussianNB, KNeighborsClassifier, RandomForestClassifier (RF), AdaBoostClassifier,and GradientBoostingClassifier. The benchmark dataset was split into the training set (75%) and the test set (25%), and then the optimal parameters of the algorithm are determined using 10-fold cross validation and a grid search strategy. The optimal model was selected by comparing the mean AUC of 10-fold cross validation under different parameter. The results show that the support vector machine (SVM) and random forest (RF) methods perform best on the benchmark dataset. Therefore, the benchmark dataset based on FP2 molecular fingerprints, and the SVM and RF methods were used in this study. We have added these contents in the revised manuscript (Line 368-381, Page 13) and in the Supplementary Materials (Supplementary Figure 2).
- D) We have investigated the probable targets of these predicted hits through molecular docking. The RNA-dependent RNA polymerase (RdRp) and 3C-like protease (3CLpro) encode by SARS-CoV-2 genome play important roles in the viral life cycle and are considered the most promising targets for drug discovery against SARS-CoV-2. We used AutoDock Vina (v1.1.2) to conduct virtual screening of 1013 active anti-SARS-CoV-2 compounds against RdRp and 3CLpro proteins, respectively. The 3D structures of the RdRp (PDB ID: 7BV2) and 3CLpro (PDB ID: 7VH8) were downloaded from the Protein Data Bank. The ligand center of the complex is used as the docking center. 3CLpr inhibitor PF-07321332 (-7.0 kcal/mol) and RdRp inhibitor Remdesivir (-5.3 kcal/mol) were selected as positive controls for RdRp and 3CLpro proteins, respectively. The docking results showed that 15 compounds specifically docked to the 3CLpro protein, 223 compounds specifically docked to the RdRp protein, and 278 compounds could dock to both the 3CLpro and RdRp proteins. Other compounds may target other viruses or host proteins (Line 294-304, Page 11, Supplementary Figure 3).
- E) The code and associated dataset have been hosted on Github (https://github.com/daishaoxing/anti-SARS-CoV-2) (Line 464-465, Page 15).
- F) Currently, virtual screening of anti-SARS-CoV-2 compounds mainly focus on the FDA approved drugs and synthetic compounds. The natural compounds from medical plants were seriously neglected. Furthermore, manly screening experiments are target-based, which target Mpro and 3CLpro proteins. Our study is based on cell phenotype data and use machine learning (RF and SVM) to screen potential natural compounds against SARS-CoV-2. The performance of our method is better than previous similar method (Lines 305-310, page 12).
In summary, we thank the reviewer’s comments again. We have integrated the above contents in the revised manuscript and performed language-editing using a paid editing service. We hope the new version will meet your standards and requirements.
Supplementary Figure 2. The AUC value of different machine learning classification methods for benchmark datasets. (a). Bar plots show the mean AUC values of 10-fold cross validation derived from different machine learning classification methods for benchmark datasets. The error bars represent the standard deviations. (b). Optimal parameters and mean AUC value of different machine learning classification methods for benchmark datasets.
Supplementary Figure 3. The pie chart showing the number of compounds against RdRp, 3CLpro and other proteins based on the docking results.
Figure 4C. The bar chart showing the number of compounds in each structure category classified by the method ClassyFire.

Round 2
Reviewer 3 Report
The revised manuscript is more precise than the previous submission but it still requires substantial correction.
The manuscript needs minor corrections in the terms of English, image and references correction.
1. Unfortunately, the authors have not included the docking information in the manuscript.
2. The image quality is very low and not understandable.
3. Protein targets have not been mentioned appropriately. The PDB ids must have present and listed in the manuscript. For the SARS-CoV-2 study, refer below-mentioned publications:
PMID: 33521709; PMID: 36423529; PMID: 33845270; PMID: 35098887; PMID: 33398013; PMID: 34526388.
4. Supplementary files are missing.
Author Response
Reviewer 3 comments and response
Comment 1: The revised manuscript is more precise than the previous submission but it still requires substantial correction. Unfortunately, the authors have not included the docking information in the manuscript.
Response: Thanks for your careful checks. As your comment, we have added a more detailed information about molecular docking.
In the Discussion section, the following contents have been added. “Furthermore, we investigated the probable targets of these predicted hits through molecular docking (see details in Material and methods) The docking results showed that 15 compounds specifically docked to the 3CLpro protein, 223 compounds specifically docked to the RdRp protein, and 278 compounds could dock to both the 3CLpro and RdRp proteins (Supplementary Figure 3, Supplementary Table 5). Other com-pounds may target other viruses or host proteins.” (Lines 317-323 in Page 11)
In the section of Material and methods, the following contents have been added. “4.5. Molecular docking
We further investigated the probable targets of these predicted hits using mo-lecular docking. The RNA-dependent RNA polymerase (RdRp) and 3C-like protease (3CLpro) encoded by the SARS-CoV-2 genome play important roles in the viral life cycle and are considered the most promising targets for drug discovery against SARS-CoV-2 [52]. The crystal 3D structures of 3CLpro (PDB ID: 7VH8) and RdRp (PDB ID: 7BV2) were downloaded from the Protein Data Bank. Water molecules, in-hibitors, and unrelated chemical complexes were removed from 7VH8 and 7BV2 us-ing Discovery Studio Visualizer (v2019). For each complex, the center of the search box was set as the center of the geometry of the crystallized ligand. Through Discov-ery Studio Visualizer, we obtained the docking centers of RdRp (center: X=91.51, Y=92.38, Z=103.73) and 3CLpro (center: X=-18.76, Y=17.14, Z=-25.14). The 1013 pre-dicted active compounds from the TCMSP were used for molecular docking. Then, the receptors and ligands were converted to PDBQT format using MGLtools (pre-pare_ligand4.py and prepare_receptor4.py, v1.5.7). Finally, docking analysis was performed using the AutoDock Vina (v1.1.2) software. The box size was set to 15 *15*15 Å, and other parameters were set to default. After docking, the 3CLpr inhibi-tor PF-07321332 (-7.0 kcal/mol) and the RdRp inhibitor Remdesivir (-5.3 kcal/mol) were selected as positive controls for the RdRp and 3CLpro proteins, respectively.” (Lines 477-494 in Page 15)
Comment 2: The image quality is very low and not understandable.
Response: We deeply appreciate your comments. As your comment, all the images have been checked and modified.
Comment 3: Protein targets have not been mentioned appropriately. The PDB ids must have present and listed in the manuscript. For the SARS-CoV-2 study, refer below-mentioned publications: PMID: 33521709; PMID: 36423529; PMID: 33845270; PMID: 35098887; PMID: 33398013; PMID: 34526388.
Response: We deeply appreciate the reviewer’s suggestion.
1) The targets information and PDB ids of the protein used for docking have been added in the revised manuscript. (Lines 479-482 in Page 15)
2) We carefully reviewed the literature you provided, and have cited these studies and about SARS-COV-2 and targets information in the revised manuscript. Please see references 34, 35, 40, 41, 42, 53.
Comment 4: Supplementary files are missing.
Response: Thanks for your careful checks. We have checked and uploaded the supplementary files.
Reviewer 4 Report
Although the authors have significantly revised the article, however, I have not found information about applicability domain of their models in the corrected manuscript.
The term applicability domain is missing from the updated manuscript.
Perhaps the authors do not understand the essence of the problem and the meaning of this term.
In this situation, I cannot recommend the article for publication.
Author Response
Reviewer 4 comments and response
Comment 1: Although the authors have significantly revised the article, however, I have not found information about applicability domain of their models in the corrected manuscript. The term applicability domain is missing from the updated manuscript.
Perhaps the authors do not understand the essence of the problem and the meaning of this term. In this situation, I cannot recommend the article for publication.
Response: We are very sorry for not understanding your meaning. Now we known that the applicability domain of model is the physico-chemical, structural, or biological space on which the training set of the model has been developed, and for which it is applicable to make predictions for new compounds. The task of the AD is to define the boundaries within which a model can be utilized and offer reliable predictions.
We carefully reviewed the literature and have supplemented the analysis of applicability domain. In the Result section, the following contents have been added. “Furthermore, we evaluated the applicability domain (AD) of the models using the pyADA [96]. The boundary of AD is determined by the max similarities of the structures present in the test set in relation to the training set (see Material and methods). The threshold of max similarity is 0.182. We found that all the 1013 predicted active compounds were inside the AD and only 53 of 5513 inactive compounds were outside the AD (Supplementary Figure 4). This indicates the relative reliability of our prediction results.” (Lines 182-188 in Page 6)
In the section of Material and methods, the following contents have been added. “4.4. Definition of Applicability Domain (AD) of Models
The applicability domain (AD) is a theoretical region in the chemical space sur-rounding both the model descriptors and modeled response. Due to the limitation of the chemical space of the training sets, it is impractical to predict a whole universe of chemicals using a single QSAR model [94]. Therefore, AD should be seriously considered in any QSAR-based predictive model. The setting of AD can prevent excessive prediction bias from the large characteristic difference between the query compound and the training set. It means that the model prediction can be considered relatively reliable if the query compound is distributed in AD; otherwise, it is less reliable. There are various methods for calculating AD of QSAR models, such as dis-tance-based methods, and probability density distribution [95]. In this study, the AD of our models is calculated by pyADA (Python Applicability Domain Analyzer). PyADA is a cheminformatics package of Python to perform AD of molecular finger-prints based on similarity calculation [96]. And the calculation of the AD consists of a scan of similarities of the structures present in the test set in relation to the training set, the best similarity threshold is the one with the lowest error and also the lowest number of molecules with similarity below the threshold. The benchmark dataset was split into the training set (accounting for 75%) and the test set (accounting for 25%), and then the similarity threshold was calculated using 10-fold cross validation and PyADA. The similarity thresholds for these 10 times are 0.14, 0.24, 0.2, 0.22, 0.22, 0.14, 0.2, 0.2, 0.06, and 0.2. And we used the mean value (0.182) as the final similarity threshold. To assess whether 6526 herb-derived compounds from the TCMSP are in-side the AD of our model, the maximum similarity of the molecular fingerprints for the query compound and all training data is computed using the Tanimoto coefficient. We found that 6473 compounds were inside the AD and only 53 were outside the AD (Supplementary Figure 4). It shows that most compounds from the TCMSP are con-sidered suitable for our models prediction.” (Lines 452-476 in Page 14-15).
Thank you again for your professional comments.
Round 3
Reviewer 4 Report
The authors finally made the necessary additions and changes.
In the last version, the paper can be published